# Comment on Bischoff et al. The Effects of the Food Additive Titanium Dioxide (E171) on Tumor Formation and Gene Expression in the Colon of a Transgenic Mouse Model for Colorectal Cancer. *Nanomaterials* 2022, *12*, 1256

**DOI:** 10.3390/nano13091551

**Published:** 2023-05-05

**Authors:** Norbert E. Kaminski, Samuel M. Cohen

**Affiliations:** 1Department of Pharmacology & Toxicology, Center for Research on Ingredient Safety, Institute for Integrative Toxicology, Rm 165G Food Safety and Toxicology Building, Michigan State University, East Lansing, MI 48824, USA; 2Department of Pathology and Microbiology, University of Nebraska Medical Center, Omaha, NE 68198, USA; scohen@unmc.edu

The publication by Bischoff et al., 2022 [1] claims to provide evidence for a promoting effect on colorectal cancer in a mouse model by titanium dioxide (TiO_2_). There are many serious flaws in this publication and the reported findings do not support the authors’ conclusion. In fact, the results presented support, just the opposite, a conclusion of no effect.

Importantly, the findings reported throughout the paper are described as not being statistically significant; therefore, how can the authors claim that there is an effect? Clearly, this is a fundamental flaw to the investigation and their discussion, which is highly speculative, has no fundamental basis in fact or data.

There are several specific issues the authors failed to address in their publication. First, in this study titanium dioxide was administered either in drinking water or in water administered by gavage and does not reflect human exposure, which is in the diet. Titanium dioxide is not water soluble, so the authors tried to fix this problem by sonicating as well as shaking the water bottles in their initial study. Sonication will not address the deficiencies of administering titanium dioxide in water, as there will be an agglomeration of the nanosized particles, and greatly distorts the size composition of the original material that would normally be present in diet. Second, the authors indicate that the E171 (i.e., food grade titanium dioxide) that they used was comprised of approximately 64% nanoparticles. Even the authors indicate in the introduction that E171 is typically comprised of 10–40% nanoparticles in number/size distribution. Even 40% nanoparticle composition would be considered unusual. Shaking the water bottles will not avoid the precipitation problem, especially if the shaking of the bottles is performed during the day, as mice are nocturnal and will eat and drink at night.

The animal model used to evaluate their hypothesis, the CAC5^Tg/Tg^; APC^5806/+^ mouse, is not an appropriate model for examining colon carcinogenesis with respect to human relevance. The tumors produced in this transgenic model are benign adenomas, not carcinomas. Likewise, there is an extensive variability in tumor incidence in studies with this model, as well as variability in the number and sizes of tumors. Background incidences can be as high as 26%, as indicated by the authors in their reference to Xue et al., 2010 [2]. The variability is obvious even in the study by Bischoff et al., 2022 [1], in which they have one of the control animals having a very large number of tumors the entire length of the colon. Based on the extensive nature of the tumors in that control animal, the animal was excluded from their analysis. Even with exclusion of that animal, the authors found no statistically significant findings in tumor incidence, tumor size, or tumor number, compared to control. Since there was no statistically significant finding, they essentially have a negative study, which supports the conclusion that titanium dioxide does not promote colorectal tumors, even in this mouse model that is highly sensitive to development of colorectal tumors.

There are several critical aspects of the pathology in this study that also raise questions. The tumors are all benign, so they should not be referred to as carcinomas. Based on the photographs provided, the tumors are adenomas or sessile serrated polyps. It is difficult to ascertain what the authors mean by hyperplasia, as the photograph that is provided illustrating such a finding is actually a sessile serrated polyp and not hyperplasia. They describe lymph nodes and lymphocytes being present and increased, but this appears to be lymphoid aggregates, a normal constituent of colonic tissue, similar to Peyer’s patches in the small intestine. That these are lymphoid aggregates is further supported by their indication that there were epithelial cells in the lymphoid tissue. If these were truly lymph nodes, this would represent a metastasis, whereas the photograph clearly indicates this is benign tissue.

The authors’ evaluation of tumor size is a meaningless parameter to evaluate, since the tumor size will be a reflection not only of the epithelial cells that are of greatest concern in the formation of these tumors, but they will also have highly variable amounts of other cell types present, such as lymphocytes, fibroblasts, macrophages, blood vessels, and even edema, which will greatly distort any relationships of size with respect to actual tumor formation.

The preparation of the materials for genomic analysis also is likely to be providing highly misleading information as evidenced by the following omissions. They do not describe what portion of the colon the tissue was taken from or if this was standardized. Various parameters in the right colon compared to the left colon can differ considerably. They also do not indicate how they validated a lack of tumor formation or other pathology already present in the tissue taken for genomic analysis. Furthermore, they do not state whether they separated the mucosa from the remainder of the colonic tissue, and especially, there appeared to have been no effort to separate epithelial cells for examination by the genomic analysis. Examining all of the cells in the colonic tissue, including mucosa, submucosa, and muscle wall, will not provide informative data with regard to carcinogenesis. The target tissue is actually the epithelium, which is a minor cellular component of the colon. In the submucosa and even in the lamina propria, there will be a large number of lymphocytes, macrophages, blood vessels, and fibroblasts, as well as the smooth muscle cells of the muscle wall. All of these would dilute any effect on epithelial cells in their genomic findings.

The analysis also has several deficiencies. The authors conducted transcriptomic analysis using microarray analysis of colonic tissue comparing control versus E171 (1, 2, and 5 mg/kg/day)-treated mice. Microarray analysis is a notoriously insensitive method for transcriptomic analysis and is rarely used in current investigations. In addition to not indicating what layers were incorporated in the tissue examined by transcriptomic analysis, they did not indicate what portion of the colon tissue was used, whether it was standardized, or whether it was verified that it was from non-tumor tissue or non-lymphoid tissue. Even by the method that they used, there was no dose or timed response, clearly indicating that the findings were related to chance and not treatment related. The greatest changes in the differentially expressed genes (DEG) were notably at the lowest E171 dose (1 mg/kg/day), a finding that is not compatible with a treatment-related effect. In Figure 7, the authors provide a heat map of all the genes differentially expressed due to E171 administered over the 21 days of treatment. Interestingly, a review of these genes fails to identify the elevation of a single gene that codes for an inflammatory cytokine or for that matter a chemokine that could account for recruitment of inflammatory cell infiltration. This is not surprising since Blevins et al., 2019 [3] were unable to demonstrate inflammation or the elevation of inflammatory cytokine/chemokine production within the GI tract, even after 100 days of treatment with E171 in the diet and even at much higher doses than used in the current study. Although no increase in the mRNA levels for any inflammatory cytokines were demonstrated by the authors, they repeatedly claim that their “pathway analysis” indicated evidence of inflammation in the GI tract, which they also claim is consistent with findings by Bettini et al., 2017 [4]. They have no basis for such a conclusion.

In the Discussion portion of the paper, the authors focus on the results of the pathway analysis, even though they had no statistically significant tumor findings. Thus, there is no factual basis for any of their conjectures described in the Discussion. They speculate on the role of IL-17 in contributing to cancer, and yet the authors provide no data indicating any change in IL-17 expression due to E171 treatment. Likewise, there is no evidence to support their emphasis on pathways related to circadian rhythm. A major shortcoming in the entire pathway analysis by these authors is the failure to discuss the specific suite of genes (i.e., more than one) in any given regulatory pathway that was found to be differentially expressed followed by the demonstration, using a functional analysis, to confirm that the pathways identified were indeed altered. A mere 1.5-fold change in the expression of a gene (the cutoff used for their transcriptomic analysis) is not evidence that a cell regulatory pathway is functionally altered.

In addition, there are several technical pieces of information not provided by these authors, such as description of what is meant by rectal prolapse, why prolapse occurred in these animals, whether the small intestine was evaluated (many of these min-related models have small intestine as well as large intestinal tumors), whether bedding was provided in the animals’ cages, and what the parental strain for the transgene was.

In summary, there are numerous concerns regarding the Bischoff et al. paper, with serious flaws in the experimental design, and a discussion based entirely on speculation with no statistically significant findings on which it is based. Collectively, the research findings, as discussed, are misleading and detrimental to a valid evaluation of risk assessment of titanium dioxide. If anything, the findings in this publication provide support for a lack of effect by titanium dioxide on colorectal tumorigenesis.

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
