# Peer review of "Comment on Bischoff et al. The Effects of the Food Additive Titanium Dioxide (E171) on Tumor Formation and Gene Expression in the Colon of a Transgenic Mouse Model for Colorectal Cancer. Nanomaterials 2022, 12, 1256"

_nanomaterials, 2023, doi:10.3390/nano13091551_

Round 1

Reviewer 1 Report

The Authors of this Comment reports a discussion on several concerns regarding flaws in the experimental design and discussion of “The Effects of the Food Additive Titanium Dioxide (C171) On Tumor Formation and Gene Expression in the Colon of a Transgenic Mouse Model for Colorectal 4 Cancer Nanomaterials, 12(8):1256, 2022”

Their comments are scientifically solid and lay on a robust discussion. In my opinion, the authors of the original manuscript have based their discussion on speculation. Indeed, they report in several place of the original manuscript that the data are not-statistically significant. On the other hand, they have strongly reduced the initial claims respect to the previous version. Overall, I feel this comment deserves to be published and would support a better understanding of the original manuscript.

Author Response

Response to Reviewer 1 Comments

Point 1: Their comments are scientifically solid and lay on robust discussion.

Response 1:  We agree with Reviewer 1

Reviewer 2 Report

The comment on the paper previously published is comprehensive. The point of view is different and will contribute for the field. The authors should be careful for not being offensive against the author’s dignity and they should take in consideration the positive aspects of the work already published even if they suggest the contrary. Finally, I am not against the publication of the author’s point of view regarding this relevant topic in science, at this 21 century.

Author Response

Response to Reviewer 2 Comments

Point 1: The point of view is different and will contribute for the field.

Response 1:  We concur with the Reviewer.

Point 2: The authors should be careful not to be offensive against the authors.

Response: The authors of the letter have carefully reviewed their letter and believe that all of the statements in the letter solely address the science presented in the paper.  We do not believe there are any offensive statement in the letter.
